# Feeding and Amines Stimulate the Growth of the Salivary Gland following Short-Term Starvation in the Black Field Cricket, *Teleogryllus commodus*

**DOI:** 10.3390/insects14060495

**Published:** 2023-05-25

**Authors:** Nurul Wahida Othman, Andrew B. Barron, Paul D. Cooper

**Affiliations:** 1Centre of Insect Systematics, Faculty of Science and Technology, Universiti Kebangsaan Malaysia, Bangi 43600, Selangor, Malaysia; wahida@ukm.edu.my; 2Department of Biological Sciences, Macquarie University, Sydney, NSW 2109, Australia; andrew.barron@mq.edu.au; 3Research School of Biology, The Australian National University, Canberra, ACT 2601, Australia

**Keywords:** periodic feeding, starvation, amine content, cell changes, insects

## Abstract

**Simple Summary:**

Biting and chewing insects, such as crickets, may not have regular meals. They have a foregut with a crop that permits food to be stored, and also for further processing of previously ingested food if they cannot find food. Does this short period of starvation cause any further changes in other parts of the digestive system? We found that the salivary glands of crickets decreased in size as a result of a decrease in the size of one type of salivary cell, the zymogen cell. Amines (serotonin and dopamine) that stimulate gland secretion appeared to be stored with those cells, rather than in the parietal cells that are involved in salivation. Upon feeding, the glands immediately increase in size and the amines are then found in the parietal cells. This work appears to show that feeding and starving can affect parts of the digestive system as shown in animals such as snakes that can go long periods without feeding. The salivary glands of insects may have several different control mechanisms that require further work.

**Abstract:**

The salivary gland of the black field cricket, *Teleogryllus commodus* Walker changed size between being starved and fed. Crickets without access to food for 72 h showed a reduction in both wet and dry mass of the glands compared with the glands from continuously fed animals at 72 h. Glands returned to size following ingestion within 10 min. Salivary glands of starved crickets (72 h) were incubated in saline containing either serotonin (5-HT) or dopamine (DA). Glands increased to pre-starvation size after 1 h incubation in situ with either 10^−4^ moles L^−1^ 5-HT or 10^−4^ moles L^−1^ DA, although lower concentrations (10^−5^ moles L^−1^) did not affect gland size. From immunohistochemistry, amines appeared to shift from zymogen cells during starvation to parietal cells following feeding. High-performance liquid chromatography showed that serotonin concentration is higher than dopamine in the salivary gland removed from starved and fed crickets, but the quantity of these compounds was not dependent upon feeding state; the amine quantities increased as gland size increased. Further work is necessary to determine what might be the stimulus for gland growth and if dopamine and serotonin play a role in the stimulation of salivary gland growth after a period of starvation.

## 1. Introduction

Periodic feeding may be associated with changes in the digestive system due to a decrease in the production of digestive enzymes and absorptive tissues [1]. Many vertebrates demonstrate a number of responses to a lack of food that includes behavior and physiological and morphological changes [2] but very little information has been reported in invertebrates. Starvation has been examined in crustaceans, especially shrimp, crabs and crayfish [3,4,5], with results indicating that biochemical, physiological and morphological changes occur in these arthropods. Longer-term periodic feeding may not be as important in many insects, although hematophagous arthropods are known to show significant changes in behavior and both physiological and morphological changes with their blood meals [6,7,8], with most blood feeders requiring a meal for growth or development of eggs.

Less is known with regard to changes associated with feeding in other insects, although the presence of a crop in many chewing herbivorous species suggests that intermittent feeding is common. As food availability may vary seasonally, the time between feeding bouts may be extended, so storage of food is necessary to permit searching and locating other food sources. Locusts have been shown to alter their digestive system morphology in relation to the amount of fiber in the food [9], and with changes in protein content [10]. However, changes in digestive system morphology that occur in response to extended periods of starvation have not been reported.

All Orthoptera have a muscular crop and meals are usually associated with filling the crop, with processing of the food through the midgut occurring over hours [11]. The crop itself varies in the internal cuticular structures that may be used in holding or moving food along the digestive system [12,13,14]. *Teleogryllus commodus* Walker (*Orthopera*, *Ensifera*, *Gryllidae*) can process food repeatedly through the crop, with food being moved back into the crop for reprocessing if further meals are restricted [15].

Along with the crop are the salivary glands necessary for moistening or initially processing ingested food. Insect orders, such as the *Orthoptera* and *Blattodea*, have lobed acinar salivary glands within the thorax with several acini joined together by tertiary ducts [16]. The glands are paired on the left and right sides of the thorax with the final duct extending to the hypopharynx and saliva is released onto the entering food. The process for producing saliva from these glands is thought to result from nervous stimulation from either the subœsophageal ganglion or via the stomatogastric nervous system in association with feeding [17], with the lack of nervous stimulation resulting in a lack of secretion. The salivary glands also may initiate digestion of carbohydrates [16], and in crickets and locusts, the salivary glands may also contribute to decreased pH in crop contents [15,18].

Serotonin and dopamine have been shown to regulate salivary gland secretion in several insect species [8,19,20,21]. These amines have been demonstrated to be present in the nerves that innervate the salivary glands as well as being distributed within the glands themselves [17,22,23]. Insect salivary glands produce a primary secretion in response to a serotonin-stimulated V-ATPase in the apical membranes of the secretory cells of the glands [24,25]. Serotonin causes some change in the transporter to increase the fluid movement from the gland into the tubular or duct system for transport into the pharyngeal part of the digestive system, possibly incorporating sodium–potassium transporters as well [26].

However, little is known regarding the morphology or the role of amines in non-active salivary glands such as might occur when food is not readily available.

We reported that the salivary glands of yellow-winged grasshoppers (*Gastrimargus musicus*) were reduced in size in insects that did not have food in the crop, but we could not distinguish whether this resulted from either not feeding or as a result of flight when the grasshoppers were caught [27]. The aim of this study is (1) to determine how starvation–feeding cycles might affect the morphology of the thoracic salivary gland of the cricket, *Teleogryllus commodus*, and (2) whether the amines serotonin or dopamine play a role in any gland changes by observing the effect of incubating semi-isolated glands of 72 h starved insects with amines, as well as measuring the concentration of these amines in starved and fed crickets. The overall aim is to determine what changes may occur in insects that demonstrate periodic feeding over days as may typically occur in field-active *Orthoptera*.

## 2. Materials and Methods

All crickets used in these experiments were bred in the laboratory under a reversed light cycle (12L:12D, lights off 7 a.m., lights on 7 p.m.) at 27 °C to ensure that cricket activity corresponded with experimental timing. Only adult males (2–4 weeks after adult moult) were used in experiments.

Seventy-two crickets were chosen randomly from the culture and weighed (two experiments of 36 crickets each). Based on body mass, animals were randomly allocated into one of six groups, so that total mass of each group was similar. Individual crickets were held in containers and given water and food ad lib (fed) while other individuals had only water available (starved) over 72 h. This experimental protocol allowed both initial and final weights to be determined for individual crickets, as well as the changes at 24 and 48 h. At each time point, 12 crickets from each treatment (fed or starved) were killed and the mass of crop and salivary gland was determined by weighing (body and crop mass, ±0.1 mg, Mettler Toledo, Melbourne, Australia, AE260 balance; salivary gland mass ±10 µg, Cahn C-30 microbalance, John Morris Scientific, Chatswood, Australia). This protocol was also used for the salivary gland length, salivary gland dry mass and protein content studies, although these studies only examined crickets after the 72 h starvation for comparison with fed crickets.

Crickets were left in the cabinet (27 °C and a reversed light cycle) in six groups and starved for 72 h. After the 72 h of deprivation, crickets were weighed (Mettler Toledo AE260) and food (lettuce and lab chow) provided (time 0) to all groups except the control. The control group of six crickets was not supplied with food, but was immediately frozen in liquid nitrogen (LN_2_). The other groups were allowed to feed and then frozen in LN_2_ at 10, 20, 40, 60 and 120 min after feeding commenced. Crickets were subsequently defrosted, re-weighed and their crops and salivary glands removed and weighed.

Microphotography (Leica MZ6, North Ryde, Australia with Nikon 4500 Coolpix camera, Rhodes, Australia) was performed while salivary glands and all digestive structures were still within the body for in situ comparison, or immediately following removal of specific tissues.

For the length of the salivary glands, the measurements were performed under the microscope (Leica MZ8, North Ryde, Australia) using a calibrated eyepiece graticule (±10 µm), following measurement of the cricket body length using a steel ruler (±1 mm). Salivary glands were removed, weighed (±1 µg, Cahn C-30 microbalance) on tared pieces of aluminum foil, and placed in a desiccator over silica gel. The desiccator with salivary glands was put in an oven at 50 °C overnight, removed and allowed to cool to room temperature prior to re-weighing the salivary glands to obtain dry mass. Protein content was assessed on a separate group of crickets starved and fed for the same amount of time. Salivary glands (both left and right sides) were quickly removed after the 72 h of treatment and placed in 200 µL cold phosphate-buffered saline (PBS) within an Eppendorf^®^, (Macquarie Park, Australia) tube. The glands were initially disrupted using a pestle, and then the tubes were placed in a sonicator bath (Unisonics Pty. Ltd., Sydney, Australia) for 5 min to complete disruption of the glands. More PBS was added to make the final volume 500 µL. Protein content was measured using a DirectDetect^TM^ analyser (Merck Australia, Sydney, Australia) with 3 × 2 µL aliquots of the solution pipetted onto the cards for measurement using an electronic pipette (Ovation Bionatural Pipette, Interpath, Somerton, Australia).

### 2.1. Histology and Cell Size

Salivary glands were dissected from the thorax and immediately fixed in cold Bouin’s fixative overnight (<12 h) in labeled vials (fed crickets with food in the crops), or starved (crickets with empty crops). Fixed glands were washed the following day with 70% ethanol and dehydrated using increasing concentrations of ethanol (70–100%). Glands were embedded in wax. Three blocks for each treatment were sectioned (7 µm), using a microtome (Leica RM5500, North Ryde, Australia). Slides with 4–5 sections were immersed in a water bath (55 °C), removed and placed on a slide tray (40 °C) (Ratek, Boronia, Australia) overnight. Sectioned tissues were then stained using the periodic acid-Schiff’s reagent (PAS) and Alcian blue method. Tissues were mounted using Depex^®^, covered with a cover slip and allowed to set overnight. Microphotography was taken using a Leica DMLB (North Ryde, Australia) with a Leica MC120HD camera. Adobe^®^ Photoshop^®^ (version CS6) (Sydney, Australia) was used to adjust levels, but all images were adjusted using the auto to prevent any bias in adjustment. Images were imported into ImageJ and cell heights were measured for both zymogen and parietal cells (Figure 1). Duct cell height was measured from duct lumen to salivary gland tissue, with only primary or secondary ducts within the tissue included. Duct diameter was also measured to determine whether secretion might change diameter. Only 1 measurement for each cell type was made per slide.

### 2.2. Incubation Studies

Eighteen crickets were weighed, ranked by mass, and subdivided within ranks into sets of 3 animals. To ensure that all experimental groups were matched for mass, from each set, one cricket was randomly allocated into one of 3 groups, until 3 groups of six animals were formed. All groups were provided with water only for 72 h and held at 27 °C on a reversed light cycle (12L:12D). The three groups were designated as control for starvation treatment (control for gland size as result of starvation), control for incubation treatment (incubation of salivary glands with cricket saline only) or crickets for incubation experiments (salivary glands incubated with serotonin or dopamine dissolved in cricket saline).

For the group that was the control for starvation, after the 72 h, each animal was weighed, anaesthetized at 4 °C for 30 min, then dissected in cricket saline to remove the salivary glands that were weighed (±1 µg, Cahn C-30 microbalance). For the experimental groups (both control and treated), the crickets were anaesthetized at 4 °C then cut opened along the dorsal surface to expose the glands. Using a dissecting microscope (Wild M8 zoom microscope) (North Ryde, Australia), petroleum jelly was used to make a small well to isolate the glands from the rest of the tissues. The glands were placed in the middle of the well and any hemolymph remaining was removed using a glass Pasteur pipette with modified narrow tips. The salivary glands were incubated with 200–300 μL (enough to cover up the whole glands) of cricket saline (control) or either 10^−4^ M serotonin (5-HT, Sigma) (Castlehill, Australia) or 10^−4^ M dopamine (DA, Sigma) dissolved in cricket saline. The glands were observed over 60 min and changes during this time noted. Images of the incubating glands were taken before and during the incubation period (Nikon Coolpix 4500) through the eyepiece of the dissecting microscope. At the end of the 60 min, salivary glands were removed and weighed (±1 µg, Cahn C-30 microbalance).

### 2.3. Quantification of Amines

The quantification of DA and 5-HT was performed using high-performance liquid chromatography (HPLC) coupled with an electrochemical detector. Twenty-four crickets were weighed, ranked by mass and separated into 3 groups, ensuring that total mass was approximately equal in each group. Crickets were starved (water available) for 72 h in the constant temperature cabinet at 27 °C under a reversed light cycle. After 72 h, one group was cooled (4 °C) for 30 min, then weighed, dissected and crops and salivary glands were removed and weighed. Food was provided for the other two groups of crickets for one or two hours. The group that had food for 1 h was weighed, dissected and crops and salivary glands removed for weighing. The crickets that had access to food for 2 h were handled in exactly the same way. Salivary glands were weighed by being placed into individual tared Eppendorf^®^ tubes then tubes containing glands re-weighed (±1 µg, Cahn electrobalance ±1 g) with salivary gland mass determined by difference. Glands were immediately frozen in LN_2_ and stored in a freezer (−70 °C) until analyzed. Dissected salivary glands were centrifuged at 15 g for three minutes at 4 °C to begin tissue breakdown. They were then homogenized in 100 µL of 200 mmol μL^−1^ perchloric acid, containing 100 pg µL^−1^ of dihydroxybenzylamine (DHBA, Sigma) as an internal standard for biogenic amine quantification. Homogenized samples were left on ice for 20 min protected from light, and then centrifuged at 15 g for 15 min to pellet cellular debris.

The biogenic amine content of 10 µL of the supernatant of each sample was analyzed using HPLC following methods in Barron and Robinson [28]. Briefly, samples were injected using an autosampler (Agilent Technologies, Mulgrave, Australia) and separated across an HR 80 column with 0.2 micron octadecylsilane packing. Biogenic amines were detected with an ESA coulometric electrochemical detector coupled to an ESA dual-channel 5014 microdialysis analytical cell (Chelmsford, MA, USA). Channel 1 of the detector was set at 425 mV for dopamine and serotonin. The mobile phase (pH = 5.6) was composed of 15% methanol, 15% acetonitrile, 1.5 mmol L^−1^ sodium dodecyl sulfate, 75 mmol L^−1^ sodium phosphate monobasic, and 5 mmol L^−1^ citric acid trisodium salt. Biogenic amine levels for serotonin and dopamine were quantified relative to the internal standard DHBA.

### 2.4. Immunohistochemistry

Thoracic salivary glands (*n* = 3 for each treatment) were fixed in Bouin’s fixative overnight (<12 h) in two different vials for either fed crickets with food in the crops and starved crickets with empty crops. The tissues were repeatedly washed with 70% ethanol to nearly eliminate the yellow color of the fixative. Tissues were dehydrated by passing through increasing ethanol concentrations (70–100%) followed by two washes in 100% Histolene^®^ (Thermo Fisher Scientific, North Ryde, Australia) and embedded in wax. Wax sections (7 µm) were produced using a rotary microtome (Leica 2122 microtome, North Ryde, Australia). Four to five sections were placed on a slide treated with gelatin and slides dried overnight. 

Slides with sections were dewaxed (Histolene^®^/Xylene substitute) and rehydrated through ethanol (70–100%). The rehydration was completed with phosphate-buffered saline (PBS). A diamond pen was used to form a well by etching each slide with a ring around sections just prior to addition of PBS to reduce section movement during incubation with solutions. Slides were blotted with wipes (Kimwipes^®^, Merck Australia, Sydney, Australia) to remove excess PBS but tissue sections were not allowed to dry.

Two drops of pre-blocking agent PBT (PBS + 0.2% bovine serum albumin + 0.1% TritonX-100, Sigma-Aldrich, Sydney, Australia) were placed on each tissue section and tissues were incubated for 20 min. The PBT was carefully removed by tilting slides and excess fluid was absorbed with Kimwipes. Tissue sections were incubated again with PBT + N (4 mL PBS + 200 µL normal goat serums) for 30 min. The PBT + N was carefully removed prior to addition of primary antibody. Tissue sections were covered with diluted primary antibody (rabbit anti-serotonin (Incstar/Abcam, lot 541317) or rabbit anti-dopamine (Sapphire/Abcam Bioscience, lot 411351)) or a negative control (PBT only). Each primary antibody was diluted 1 in 1000 (1 µL in 1 mL of PBT + normal goat serum). To test the dopamine antibody reaction, the same protocol used for cockroach salivary glands Baumann et al. [29] was followed. Slides were incubated at 4 °C overnight in a sealed incubation chamber humidified by lining the base with a damp paper towel. The following day, slides were removed from the incubation chamber and repeatedly rinsed with PBS to wash off excess serum.

For secondary antibody (anti-rabbit IgG conjugated with FITC (Serotec/Sigma 011008)), tissues were incubated with PBT plus normal goat serum for 20 min. Next, the tissues were covered with secondary antibody, diluted 1:300 in PBT plus normal goat serum. The secondary antibody was allowed to remain on the tissues at 4 °C for 24 h. After the incubation period, excess antiserum was washed from the tissue by multiple rinses with PBT, followed by washing with PBS. The slides were dehydrated through an ethanol series and mounted using Sub-X mounting media. The slides were placed on a slide warmer overnight. Images were captured using a confocal laser scanning microscope (LSM 5 Pascal Exciter, Zeiss, Macquarie Park, Australia) and converted using Zen 2008 (Zeiss Confocal, Macquarie Park, Australia).

Whole glands were removed from fed and starved crickets and immediately fixed in freshly made paraformaldehyde (4%) for 30 min at room temperature. Ten washes using PBS was used to remove the paraformaldehyde, and made permeable by using methanol (5 min in 70% MeOH-PBS, 60 min in 100% MeOH, 5 min in 70% MeOH-PBS) followed by 10 washes in PBS to remove any remaining methanol. Glands were washed in PBT (PBS with 0.2% BSA and 0.1% Triton X-100), incubated for 30 min in PBT + N (4 mL PBT + 200 µL normal goat serum) and then incubated in primary antibody diluted 1:2000 (rabbit anti-serotonin (Incstar/Abcam, lot 541317) in PBT + N overnight at 4 °C. The primary antibody was removed by washing in PBT, and then incubated in 100 µL PBT + N for 30 min. The tissues were then incubated overnight in the secondary antibody (DyLight549-conjugated Affini-Pure goat anti-rabbit IgG (H + L) (Jackson ImmunoResearch)) (1:300). Tissues were finally washed in PBT, followed by PBS and mounted on slides in 70% glycerol-PBS. Micrographs were taken on a Leica DM5500B fluorescent wide field microscope (North Ryde, Australia) with a Leica 7000T camera. Optical sections (3 µm) were taken on the tissue in both differential interference contrast (DIC) and fluorescent mode. Individual lobes of the glands were reconstructed from the optical sections using the Leica software followed by using the overlay software to place the fluorescent image onto the DIC image. Fluorescent images were reconstructed using the maximum external focus setting.

All chemicals used were obtained from Sigma-Aldrich Australia. Cricket saline was made up in 500 mL and consisted of (g/500 mL) NaCl 3.36, KCl 0.45, MgCl_2_ · 6H_2_O 0.406, NaH_2_PO_4_ · 2H_2_O 0.312, glucose 0.9, sucrose 13.6 dissolved in reverse osmosis water. Then, 1.75 mL of 1 mol L^−1^ CaCl_2_ was added while stirring to prevent precipitation. The solution was then made to pH 7.15 using 1 mol L^−1^ NaOH. The serotonin and dopamine were made from stock 5 mmol L^−1^ solutions by adding the cricket saline to stock solution to dilute the stock to the final concentration.

### 2.5. Statistics

Statistical analyses were performed using JMP (version 13) or R studio (version 1.1.453). Data were analyzed using analysis of covariance with either crop mass or body mass as the covariate, or as simple analysis of variance (cell and duct measurements). For analysis of the quantity of amines, salivary gland mass was the covariate. Non-significant interactions between covariate and treatment were removed from models. Significance of a model is assumed if *p* < 0.05. All pairs of treatments were examined post hoc using Tukey’s HSD or *t*-tests with level of error at 0.05. Length of glands relative to body length and gland water content for fed and starved crickets were compared using the non-parametric Wilcoxon test, as these measurements were percentages.

## 3. Results

### 3.1. Changes by Starvation

Initial weights for *T. commodus* ranged between 0.60 and 0.80 g. Body mass decreased slightly over the three days for both fed and starved groups, but a larger decrease occurred in the starved group (Table 1). The ANOVA showed that both time (F_[2,66]_ = 3.41, *p* = 0.04) and treatment (F_[1,66]_ = 7.51, *p* = 0.009) were significant factors in explaining changes in body mass between the start of the feeding studies and when insects were killed and dissected. There was no significant interaction term (F_[2,66]_ = 0.39, *p* = 0.68).

There was a significant difference in crop mass between the fed and starved crickets (Table 1), although time of starvation did not affect the difference. Crop mass was significantly related to the body mass of crickets at time of dissection (F_[1,36]_ = 4.27, *p* = 0.042), and there was a significant interaction term between the body mass and whether animals were fed (F_[1,36]_ = 7.60, *p* = 0.007), with crop mass being maintained in fed animals relative to body mass, but a reduction in crop mass relative to body mass in starved crickets. Although crop mass decreased over time both for the fed groups and starved groups, the decrease was not significant and not included in the statistical model; however, there was a slight increase in crop mass at 72 h compared with 48 h in starved crickets (Table 1). Surprisingly, not all crickets that had access to food had food in their crop when autopsied, but their body mass was maintained during the experimental period, so they were included as fed crickets in these analyses.

Salivary gland mass did not differ between fed and starved crickets at 24 or 48 h, but was significantly different after 72 h of starvation (Table 1). Salivary gland mass significantly increased with crop mass (F_[1,36]_ = 11.69, *p* = 0.001), and therefore glands were heavier for fed than starved crickets. Salivary gland mass significantly decreased over the time of the experiment (F_[2,36]_ = 8.15, *p* = 0.001), but to a much greater extent in starved crickets compared to fed crickets (time*treatment interaction, F_[2,36]_ = 3.32, *p* = 0.041).

The length of the salivary glands in the thorax was significantly reduced in starved crickets compared with fed crickets (Table 2), although the glands were still anchored by various tracheal connections. The proportion of the body length in which the salivary glands extended was also reduced in the starved crickets compared with the fed animals (Fed 28.6 ± 0.89% (*n* = 7); starved 25.1 ± 0.97% (*n* = 6); Wilcoxon χ^2^_[1]_ = 5.22; *p* = 0.02) (Figure 2). Corresponding to the change in length, the dry mass of the glands was also reduced in crickets deprived of food for 72 h, compared with fed insects (Table 2) and despite the change in mass of the salivary glands, no difference in protein content was found (Table 2). Water content as a percent of the glands (Table 2) was not different between the treatments (Wilcoxon χ2[1] = 0.14, *p* > 0.5).

Measuring cell heights for the three different cell types of the salivary glands (zymogenic, parietal and duct cells) indicated that starvation led to the smallest zymogen cells and that crickets that had food in the crop had the largest zymogen cells (Figure 3A,B, Table 3, t ratio = 16.37, *p* < 0.0001). Parietal and duct cells did not change with feeding (t ratio > 0.15, *p* > 0.5) and no difference was found for duct diameters with feeding (F = 0.30, *p* = 0.74).

### 3.2. Changes in Salivary Glands following Feeding

Crops with food and salivary glands were larger than crops and salivary glands from starved crickets (Figure 4A). The mass of crops increased within ten minutes of feeding (Figure 4B) and reached a plateau of around 60 mg after 1 h. The increase in mass of the crop occurred only as a result of the time of food availability (F_[5,22]_ = 6.54, *p* = 0.0007), as body mass was not a significant covariate.

Salivary gland mass significantly increased in 10 min (F_[5,21]_ = 5.58, *p* = 0.002), but no further increase in mass was observed for the duration of the experiment (Table 1). Crop mass was not a significant covariate, but body mass was a significant covariate variable for salivary gland mass (F_[1,21]_ = 12.57, *p* = 0.005). No interaction terms were significant.

### 3.3. Incubation of Glands from Starved Crickets with Amines

Incubation of salivary glands in situ with serotonin and dopamine indicated that glands significantly increased in mass (F_[3,29]_ = 57.04, *p* < 0.0001). No difference for gland increase was observed between incubation with serotonin or dopamine (Figure 5). The saline-incubated glands were the same mass as the glands of crickets just starved for 72 h and removed, but not incubated (mean ± S.E. (*n*); 3.3 ± 0.27 (12) vs. 3.2 ± 0.31 (10) mg).

### 3.4. Quantification of Serotonin and Dopamine by HPLC

The amount of serotonin and dopamine varied from 462.6 to 1371.6 pmol mg^−1^ for serotonin and 250.9–923.7 pmol mg^−1^ for dopamine, with the lower amounts associated with starvation. The amount of serotonin is higher than the quantity of dopamine in the salivary gland of *T. commodus*. As the mass of glands increased, so did the amount of amine (F_[1,34]_ = 18, *p* < 0.0002), and the serotonin was significantly higher in content than dopamine (F_[1,34]_ = 38, *p* < 0.0001) (Figure 6). Including whether crickets were fed or not in the model was not significant, so it was omitted from the analysis.

### 3.5. Immunohistochemistry

Serotonin and dopamine were present in the sections of salivary glands of both fed and starved animals with the only differences in staining location (Figure 7), as immunostaining in glands from starved crickets were localized to zymogen cells, compared with glands from fed crickets where staining occurred around the parietal cells. In the whole gland images, the separation in staining for serotonin is clearly shown in the overlays, as the fed animals have clear points of staining on the periphery, in the parietal cells, but the starved animals have no staining at all in the periphery, but staining throughout the central part of the salivary lobe (Figure 8).

## 4. Discussion

The body mass of crickets decreased over time when crickets were starved, but not when food was available. Clearly, the crickets were using their nutrient reserves to meet their energy requirements during starvation. The European cricket, *Gryllus bimaculatus*, has been shown to maintain its metabolic rate over 7 days of starvation, but to switch from using carbohydrates to lipids to support this energy expenditure [30]. When dissecting the crickets in this study, the fat body appeared to have decreased in starved crickets compared with the fed animals (unpublished), suggesting that the fat is being metabolized over 3 days of starvation in *T. commodus*.

The crop mass varied with food availability, demonstrating that the crop was used to store food. The crop mass of a cricket starved for 3 days was slightly heavier than crops from crickets starved 24 h. The increase in mass with time indicates that food is re-distributed from the other parts of the digestive system back into the crop [15,31].

The change in mass, length and dry mass of the salivary glands demonstrates that crickets can reduce the activity of the salivary glands in association with food intake. The changes may be associated with the fluid movements into the digestive system to mix with the food, and glands that are actively secreting fluid are likely to be heavier than non-secreting glands, but that is not a result of a large change in water content (Table 2). Surprisingly, the glands from the starved crickets also have a reduced dry mass, suggesting that a component of the dry mass is also reduced in non-secreting glands. The reduction could be tissue or cell contents, such as any enzymes that may be synthesized and secreted by the glands, and this could be associated with the decreased size of zymogen cells. Salivary glands of crickets have not been reported to produce enzymes in large quantities [12], although the zymogen cells of locusts have been suggested to store compounds within the vacuoles that may be enzymes or other proteins involved in digestion [32]. Amylase has been reported to be present at significant levels within the salivary system of crickets and cockroaches [12,33], while locusts may have trypsin- and chymotrypsin-like enzymes present as well [11]. A reduction in the synthesis of these enzymes following their release after the last meal could account for the reduction in dry mass and zymogen cell size, but other components that are present, such as mucin, within the zymogen cells could also occur. The change in location of staining of the amines with starvation does suggest that part of the variation is a shift in amines, with zymogen cells storing amines when gland secretion is reduced. As protein does not change in salivary glands between fed and starved crickets, protein per se is not affected by starvation, but the change in amines may result from excess amines being absorbed by vacuoles within the glands.

Presumably, the salivary glands shrink because of the lack of feeding activity that stimulates the gland to produce saliva. However, the mechanisms for maintaining the gland in fed insects is unknown, although the release of amines from the zymogen glands may be part of the initiation of the salivary flow following the signal that feeding is occurring. The signaling system for initiating gland activity may be from the subesophageal ganglion, or the stomatogastric nervous system, as crickets have a lobe of the salivary gland associated with the retrocerebral complex [34].

The presence of food in the crop appears to determine if the salivary glands are active in the cricket. In *Rhodnius prolixus*, the calcium-dependent apyrase activity was absent in the crop if the salivary gland was removed [35], and presumably other enzymatic activity may be affected either directly or indirectly if salivary glands are inactive. Either the ingestion of food or even the odor might trigger the salivary glands to initiate secretion [36,37]. Certainly, the salivary secretions are influenced by a number of amines and peptides (reviewed by Ali [17]), and the absence of these stimulators may be what causes the reduction in the size of the salivary gland.

Salivary glands increased in size within 10 min of food availability, although the crop continued to increase in size until 60 min. Crop mass may be an explanatory variable in determining the mass of the salivary glands, but the stimulus for the swelling is not specifically determined by the crop mass. More likely, the feeding process itself initiates the increase in salivary gland mass. Feeding in insects is known to affect both nervous and endocrine pathways [38,39,40], therefore the direct stimulus for the increase in salivary gland size is unknown, but the movement of serotonin into the zymogen cells and away from the parietal cells with starvation may be reversed when feeding commences.

Incubation of the glands from starved insects does show that both serotonin and dopamine are capable of increasing the size of the gland, but the concentration and time course for swelling does not replicate what is seen during the feeding study. The glands only swell after about an hour when exposed to 10^−4^ moles L^−1^, as lower concentrations and shorter time periods did not stimulate the glands to the same extent as the in vivo feeding. The effective dose of the amines in the cricket glands is unknown, and if part of the stimulus is the release of amines from the zymogen cells, then local concentrations may be much greater. The concentrations of amines administered in studies of isolated salivary glands of cockroaches to initiate secretion is between 10^−9^ and 10^−4^ moles L^−1^ [19], and an increase in cAMP occurred in locust salivary glands between 10^−8^ and 10^−5^ moles L^−1^ [41]. If the salivary glands of *T. commodus* are stimulated to secrete at these concentrations, then the increase in gland size found in this study is not simply a result of fluid secretion, although that may be a component of the swelling. The rapid transfer of serotonin from zymogen to parietal cells could account for the difference in time of response between feeding and incubation time courses.

The HPLC studies showed that more serotonin than dopamine is present at any mass of salivary gland. No significant difference between starved and fed crickets was found for either amine when gland mass was included as a covariate. The difference in quantity between serotonin and dopamine was also reported for locusts [41,42] although the conditions for measurement appeared to only compare feeding animals in the earlier papers. Fed and starved crickets did not significantly differ in salivary gland amine content suggests that only gland swelling following feeding accounted for apparent differences in amines. However, the distribution of the amines did change and that might be a signal that is necessary to initiate the changes in parietal membrane pumps, such as the H^+^-ATPase [19,26], that has been suggested to be involve in salivary gland fluid production, presumably as that enzyme may only be associated with the parietal cells.

Despite these observations, the conclusion is that serotonin and dopamine do increase in quantity with the increase in salivary glands. The change in amine content implies that amines are produced as the glands become more active and that the location of the amines shifts from the zymogen to parietal cells. Some other compound may be involved with the initiation of the increase in gland size in vivo that then permits the serotonin and dopamine to cause salivation, especially as the quantity of exogenous amine necessary to increase gland size was only slightly greater (100 µmoles L^−1^) than the quantity measured using HPLC (~20 µmoles L^−1^), but the time course of swelling was much longer in the bath than in vivo. The change in serotonin location from zymogen to parietal cells may be either local transport or some paracrine activity that is currently unknown. Mosquitoes have recently been shown to have a number of changes in their transcriptome related to amines controlling salivary glands [43], but few other studies have examined how starvation may influence the regulation of salivary gland function.

Future work needs to determine what compounds may be involved, but the peptides that have been demonstrated to influence salivary secretion (e.g., SchistoFLRFamide or adipokinetic hormone) [44,45] may be good candidates for initial examination.

## 5. Conclusions

Short periods of starvation result in a change in salivary gland size and cells. However, the changes are rapidly changed once feeding occurs, suggesting that there is a switch in salivary gland activity. Amines appear to be part of the story in causing the change in salivary gland activity, but other chemicals may be involved as suggested previously [17]. A model (Figure 9) of how the glands may be controlled by components of the stomatogastric nervous system (SNS) (retrocerebral complex + lobe of gland adjacent [34]) that interact with feeding to control the activity of the glands. The stimulus of the SNS is necessary in addition to feeding to cause glands to be active, as our work suggests that although the crop has a role, if the cricket regurgitates food from the mid- or hindgut back into the crop, the glands are still inactive.

## Figures and Tables

**Figure 1 insects-14-00495-f001:**
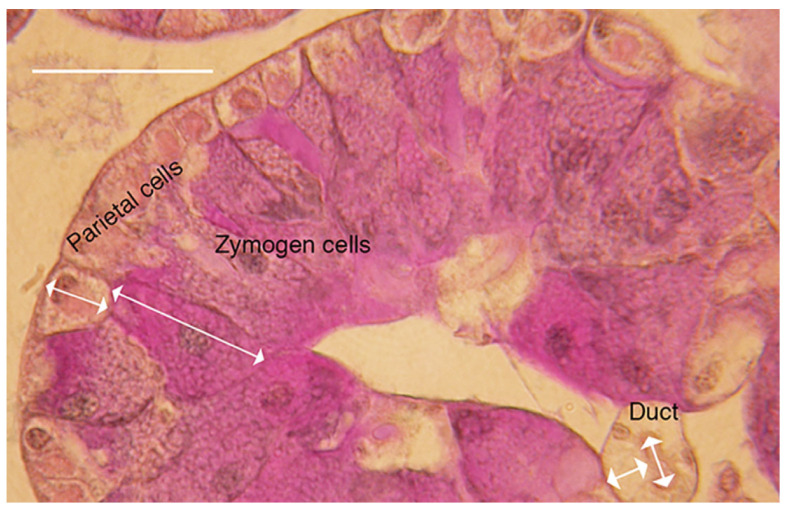
Lobe of the cricket salivary gland showing measurements (double arrows). Parietal cells are on the serosal margin, zymogen cells are darkly stained and inside the parietal cells and a secondary duct is shown. Double arrows for heights of parietal, zymogen and duct cells and lumen show location of measurements. Bar = 50 µm.

**Figure 2 insects-14-00495-f002:**
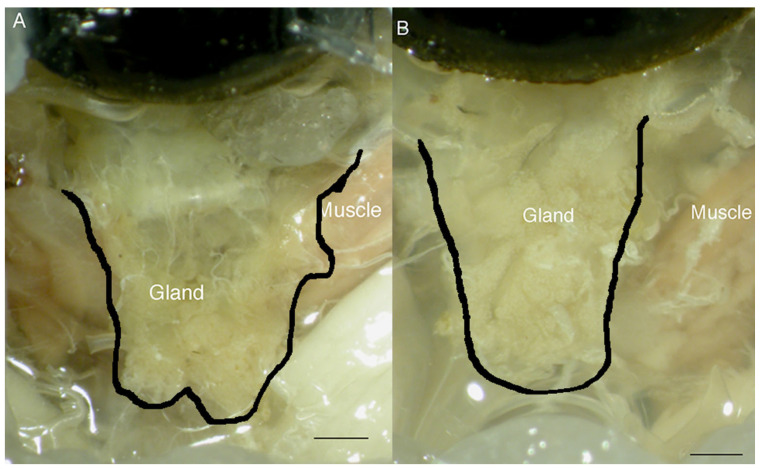
Comparison of salivary glands in fed (**A**) with starved (**B**) cricket showing the difference in whole gland size. Glands are outlined using the darkline. Scale bar = 500 µm.

**Figure 3 insects-14-00495-f003:**
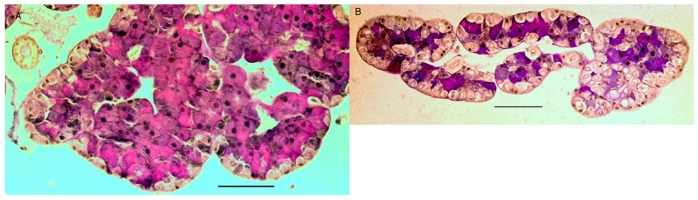
Transverse section of salivary gland removed from. (**A**) Fed crickets showing large zymogen cells. (**B**) Starved crickets (72 h) showing zymogen cells that appear smaller and less stained. Scale bars = 100 μm.

**Figure 4 insects-14-00495-f004:**
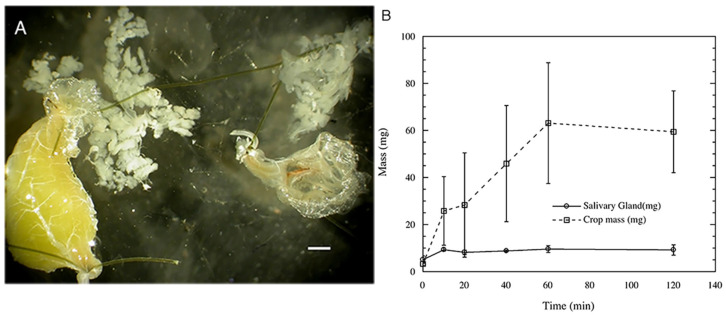
Effects of starvation and feeding on crops and salivary glands. (**A**) Comparison of crops of fed cricket (**left**) and starved (**right**) after 72 h. The salivary glands are present just above each crop. Scale bar = 1 mm. (**B**) Effects of feeding on crop and salivary gland mean mass with time. Although the salivary gland mass increases to maximum after 10 min of feeding, crop mass is still increasing over 1 h.

**Figure 5 insects-14-00495-f005:**
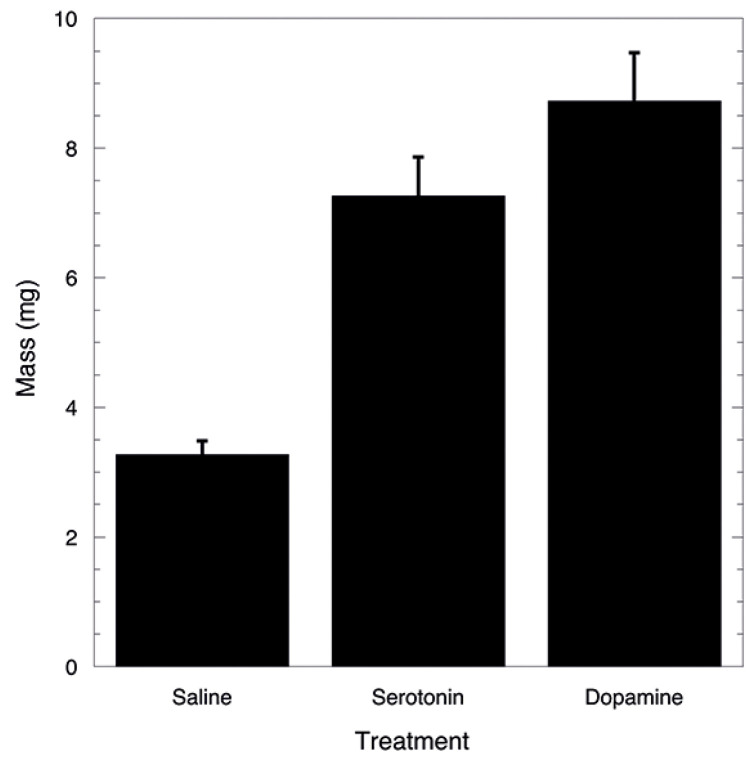
Comparison of mass in salivary glands following incubation with serotonin or dopamine dissolved in cricket saline. Saline indicates that only saline was present in the incubating bath and was the control. Both serotonin and dopamine (10^−4^ mole L^−1^) significantly increase mass of the incubating glands within 1 h.

**Figure 6 insects-14-00495-f006:**
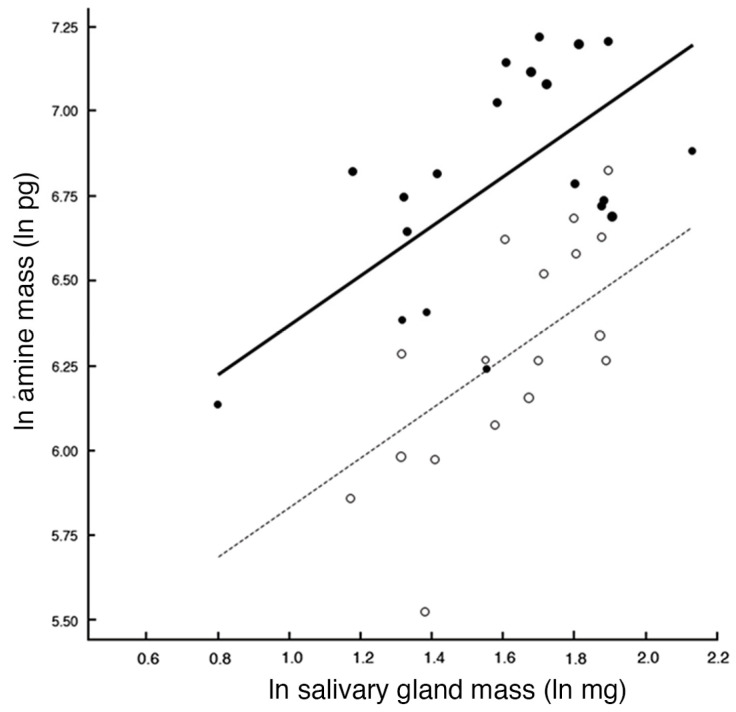
Analysis of covariance of quantity of amine (in pg) relative to mass of salivary gland (in mg). Because of non-linear relationship, data were transformed to natural logs. Both serotonin and dopamine increased with mass of the salivary gland, but there was no effect of feeding on either amine. The increase in amine is simply related to the increase in gland mass with feeding. Solid line and filled circles indicate serotonin; dotted line and open circles indicate dopamine.

**Figure 7 insects-14-00495-f007:**
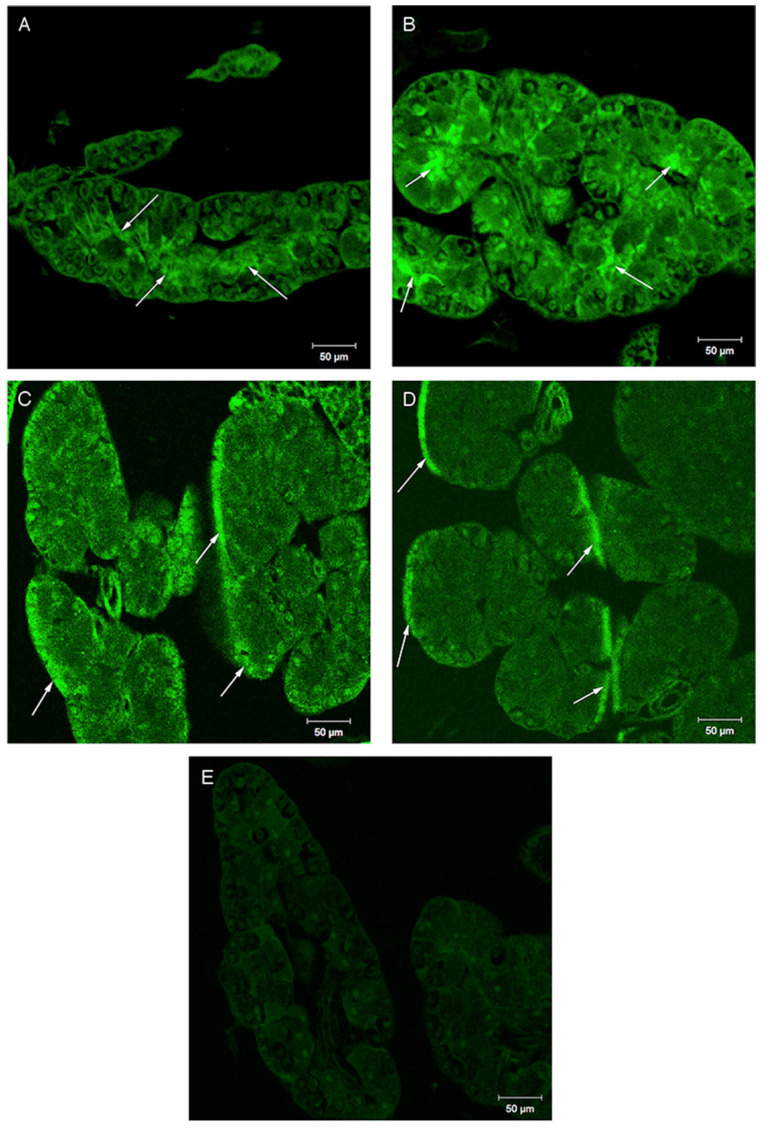
Confocal images of tissue sections stained for serotonin (**A**,**C**) or dopamine (**B**,**D**). Images (**A**) and (**B**) are thoracic salivary glands from starved crickets, while (**C**) and (**D**) are glands from fed crickets. Image (**E**) is a control section where the primary antibody was omitted from the staining procedure. Arrows indicate areas of high staining.

**Figure 8 insects-14-00495-f008:**
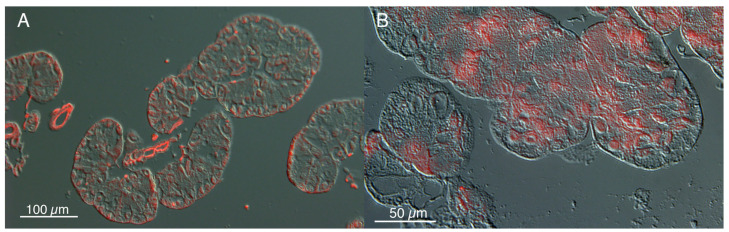
Comparison of location of cells in fed (**A**) and starved (**B**) showing serotonin immunoreactivty in overlays of maximum focus reconstructions onto differential interference contrast reconstructions using optical 3 µm sections for each. Higher magnification is presented in (**B**) so that exclusion of parietal cells is clear. Images were captured with a Leica DM5500B fluorescent microscope connected to a Leica T7000 camera. Leica software was used for reconstruction and overlay.

**Figure 9 insects-14-00495-f009:**
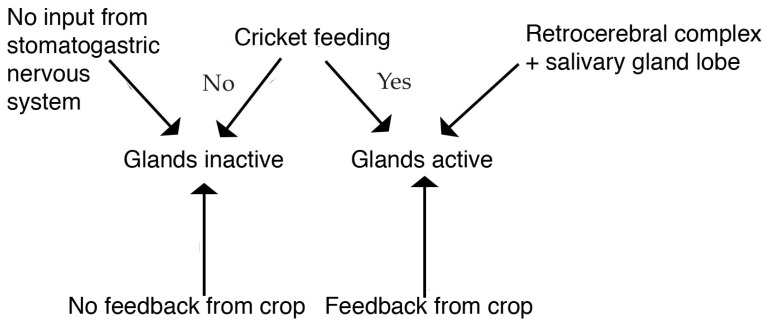
Model suggesting that stomatogastric nervous system and crop interact with feeding to control activity of thoracic salivary glands.

**Table 1 insects-14-00495-t001:** Mean mass change for fed and starved crickets at 24, 48 and 72 h. Crop mass in relation to time of feeding and treatment (fed or starved) with body mass as a covariate. Least squares mean ± S.E. Mass of salivary glands in relation to time and treatment (fed or starved) with crop mass as a covariate. Least squares mean ± S.E. *n* is shown on each group or next to the mean. Superscript letters indicate differences across treatments.

Time (h)	Fed (*n* = 18)	Starved (*n* = 18)
	Change in body mass (g)	
24	0.065 ± 0.012 ^a^	0.022 ± 0.012 ^ab^
48	0.048 ± 0.012 ^ad^	0.005 ± 0.012 ^bd^
72	0.044 ± 0.012 ^a^	-0.025 ± 0.012 ^b^

	*n* = 15	*n* = 15
	Crop mass (mg)	
24	17.1 ± 1.96 ^a^	5.7 ± 1.96 ^b^
48	13.8 ± 1.96 ^a^	5.7 ± 2.03 ^b^ (*n* = 14)
72	12.6 ± 1.96 ^a^	9.1 ± 1.96 ^a^

	*n* = 15	*n* = 15
	Mass of salivary gland (mg)	
24	5.9 ± 0.37 ^a^	6.1 ± 0.36 ^a^
48	5.5 ± 0.35 ^a^	4.6 ± 0.38 ^ab^ (*n* = 14)
72	5.4 ± 0.35 ^a^	3.8 ± 0.35 ^b^

**Table 2 insects-14-00495-t002:** Comparison of changes in length of salivary glands (emergence from pharynx to last acini in thorax), dry mass of salivary glands, protein content and water content between starved and fed (72 h) crickets. Mean ± S.E. (*n*). Water content was compared with the Wilcoxon test.

	Fed	Starved	F Ratio	*p*-Value
Length of salivary glands (mm)	7.4 ± 0.15 (7)	6.4 ± 0.16 (6)	21.4	0.007
Dry mass of salivary glands (mg)	0.73 ± 0.06 (17)	0.53 ± 0.06 (15)	6.16	0.02
Protein content (mg/mL)	1.6 ± 0.17 (9)	1.4 ± 0.18 (8)	0.86	0.37
Water content (%)	85 ± 0.01 (17)	86 ± 0.01 (15)	NS	

**Table 3 insects-14-00495-t003:** Comparison of heights (µm) of cell types (zymogen, parietal, duct) from the salivary glands removed from crickets either fed or starved. Mean ± S.E. *n* = 10 measurements for each cell type with slides selected randomly from each treatment cricket. Duct diameter is the distance between cells through duct opening. Different superscript letters indicate statistical differences with treatment for cell type.

Cell Type/Treatment	Fed (µm)	Starved (µm)
Zymogen	33.6 ± 0.65 ^a^	19.1 ± 0.65 ^b^
Parietal	19 ± 0.65	18.8 ± 0.65
Duct	16.3 ± 0.65	16.0 ± 0.65
Duct lumen diameter	44.9 ± 1.82	44.6 ± 1.82

## Data Availability

Data are available from the Data Commons of the Australian National University by request.

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
