# Peer review of "Feeding and Amines Stimulate the Growth of the Salivary Gland following Short-Term Starvation in the Black Field Cricket, Teleogryllus commodus"

_insects, 2023, doi:10.3390/insects14060495_

Round 1
Reviewer 1 Report
Dear Authors,
I have had the opportunity to review the manuscript titled "Feeding and amines stimulate the growth of the salivary gland 2 following short-term starvation in the black field cricket, Teleogryllus commodus ." The study is intriguing and the authors have presented a well-detailed Materials and Methods section outlining the experimental design, sample preparation, and data analysis procedures. However, I recommend that the authors revise the section with separate paragraphs and subtitles to provide specific information on the tests and assays performed.
Additionally, it is unclear whether statistical tests were performed to determine the significance of the results obtained, and what statistical model was used. Therefore, I suggest that the authors provide this information and explain the choice of statistical model. This would enhance the transparency and reliability of the study.
Furthermore, the description of microscopy techniques used is inadequate. It does not explain how the samples were prepared, what type of microscopy was used, or how the images were analyzed. Hence, I suggest the authors revise this section to provide a more detailed explanation.
Overall, the manuscript has the potential to contribute significantly to the field of entomology, and I recommend that the authors make the necessary revisions to enhance the clarity and transparency of the study. I look forward to reviewing the revised manuscript.
Sincerely
Overall, the English language used in the above text is fairly good. The writing style is scientific and technical, which is appropriate for a research article. The sentences are generally well-formed and free of grammatical errors.
However, there are a few instances where the language could be improved for clarity. For example, in sentence 280, it would be clearer to say "not all crickets that had access to food had food in their crop" instead of "not all crickets that had food had food in the crop." Additionally, in sentence 296, it would be clearer to say "in starved crickets compared to fed crickets" instead of "in starved crickets compared with fed crickets."
There are also a few instances where the language is somewhat awkward or confusing. For example, in sentence 285, the phrase "Part of the observed change resulted as" is not a clear or concise way to express the relationship between salivary gland mass and crop mass. Similarly, in sentence 297, the phrase "the glands were still anchored within thoracic tissue by various tracheal connections" is somewhat convoluted and could be expressed more simply.
Overall, while the quality of English language in the text is good, there is still some room for improvement in terms of clarity and concision.
Author Response
Subheadings were added to the methods to clarify the various topics. The preferred statistical method was an ANOVA and F tests were usually calculated using the program JMP or as pointed out, if results were percentages, then non-parametric Wilcoxon test was used. I tried to clarify the methods used for tissues for histology and for immunohistology to a greater extent, as well as the microscopes used for each component of the research.
Reviewer 2 Report
The manuscript submitted by Othman and co-workers presents an interesting study of the modulation of the size and activity of salivary glands in crickets, depending on the nutritional state of the insects. Combining classical and modern approaches, the authors describe the dynamics of changes in salivary glands induced by starvation and by feeding, correlating these changes with the levels of two biogenic amines.
In the opinion of this reviewer, the study is well-conceived and the findings are interesting. Yet, some aspects could be improved for clarity.
Specific points:
- line 30: an space is missing between "concentration is" and "higher"
- lines 174-175, it seems that the space between the two lines is too big, please check
- lines 195-196, same as previous
- lines 280-282. The sentence is a bit confusing. Is the fact that body weight has not changed that made empty-crop fed crickets be considered as "fed"? Please, clarify.
- Figure 2. The figure is too small and the inserted legends not contrasting enough against the background. Maybe those glands could be delimited with a dotted line.
- Figures 4, 5, and 6. They are too small. Please increase the size.
- Discussion. Several passages should be improved. The first sentence of the second paragraph is a general fact about most insects. The second sentence of the same paragraph is confusing; where does food come from? The paragraph in lines 426-433 condenses a lot of information, please, explain each assessment properly. Lines 469-470; the first sentence of this paragraph needs to be reformulated in a more logical manner. For instance, "... actual or potential feeding triggers the release of biogenic amines that stimulate, among other targets, the activity of salivary glands."
Additional points:
- were the levels of dopamine and serotonin quantified before and after feeding in the hemolymph or only in salivary glands?
- it would be nice adding a schematic flow diagram linking factors, processes, target organs, and physiological/morphological consequences.
Author Response
All minor points were addressed as indicated. I tried to clarify the parts requested. We have added Figure 9 to summarise the control of the salivary gland activity, assuming that gland growth with feeding indicated preparation for an increase secretion of fluid. All figures were increased in size and Figure 2 had a line added to indicate the extent of the salivary gland in the cricket body.
Reviewer 3 Report
Othman et al. (Feeding and amines stimulate the growth of the salivary gland following short-term starvation in the black field cricket, Teleogryllus commodus) report that the effects of hunger in the morphology of the salivary glands and how the biogenic amines (5-HT and dopamine) involves in this process in the black field cricket Teleogryllus commodus.
Overall, the paper is not well-written. Especially the materials and methods section is redundant and not well organized. In some parts, the authors described too much information in detail, but in others, necessary information is missing or described in an inappropriate part of the manuscript. Some technical terms are inconsistent throughout the manuscript. I really struggled to understand what the authors did. This manuscript must be reorganized before publication. In addition, the quality of the photographs is low. The conclusion of the paper largely relay on histological analyses; therefore, the authors must present better photographs with appropriate annotations.
Despite the above problem, measurements of the size of tissues and HPLC analysis to measure the biogenic amine contents seems OK. However, I have severe concerns with the results and conclusion of the immunostaining (Figure 7). The insect salivary glands receive aminergic innervation from the CNS, which regulates the activity of the glands. Therefore, the 5-HT/DA immunoreactivity localizes at the axon terminals of the aminergic innervations. Thus, the uneven distribution of the immunoreactivity in the tissue sections is due to the non-specific/uneven binding of the primary/secondary antibodies rather than the starvation-dependent shift of the location of the amines from the zymogen to parietal cells. I strongly recommend conducting whole-mount immunostaining and confocal microscopy to carefully observe the localization of 5-HT/DA and the structural changes caused by starvation in the salivary glands.
As mentioned above, the paper is not well-written. This manuscript must be reorganized before publication.
Author Response
The methods were partially re-written, but the subheadings added to methods help to isolate the various parts of the work to avoid redundancy of methods. To clarify the section immunohistochemistry, we added whole lobe components where DIC was used to show the unstained tissue with an overlay of the stained tissued was used to show the separation of staining for serotonin within a lobe when comparing fed and starved crickets (Figure 8). All figures were increased in size with 300 dpi to help clarify the presented images.
Round 2
Reviewer 1 Report
I am pleased to see that the authors have put in significant effort in the revised version of the resubmitted manuscript. They have effectively addressed the issues and concerns that were previously raised. As such, I accept this manuscript in its current form.